# The impact of long-term care insurance on household expenditures of the elderly: Evidence from China

**Tianshu Zhang**⬛*, **Zeping Hu, Kaiyue Zhang, Xinran Li**

School of Public Administration, Inner Mongolia University of Finance and Economics, Hohhot, China

* zhangtianshu314@126.com

## Abstract

This study aims to investigate the impact of China's long-term care insurance (LTCI) pilot on household expenditures of the elderly. Utilizing the China Health and Retirement Longitudinal Study (CHARLS) 2015–2020 three-period longitudinal panel data, we examine the policy effects of LTCI using the Differences-in-Differences (DID) approach. The results indicate that the implementation of LTCI significantly reduces medical ($p<0.05$) and healthcare expenditures ($p<0.05$) for elderly households, while substantially increasing non-medical healthcare expenditures ($p<0.01$) and total expenditures ($p<0.01$). This effect is more pronounced for older households in rural areas or with lower levels of education. Furthermore, the improvement in household expenditures is strongly associated with the health status of the elderly and intergenerational economic support. These findings provide empirical evidence that LTCI enhances household expenditures and the quality of life for the elderly, which is crucial for the development of LTCI in China and other middle-income developing countries.

## 1. Introduction

Changes in the age structure of the population and increased average life expectancy have led to a significant deepening of global population aging [1]. By 2050, the number of people aged 65 and over is expected to rise to 1.5 billion, constituting one-sixth of the total global population. China, among the many countries facing severe challenges due to population aging, will have approximately 216 million people aged 65 and above by the end of 2023, accounting for 15.4% of its total population [2]. This marks China's transition to a senior aging society just 23 years after it first entered an aging society in 2000. Furthermore, it is predicted that by 2050, the number of people aged 65 and above in China will reach a staggering 366 million. Aging presents both a constraint on economic development and a source of various family problems [3,4]. As individuals age, their physical functions decline, which significantly increases the likelihood of suffering from chronic diseases. Studies indicate that the prevalence of chronic diseases among elderly individuals reaches approximately 56% [5]. On average, these individuals live with chronic diseases for about 8.7 years, leading to high rates of incapacity and an urgent need for long-term care (LTC) services [6]. In 2019, the average total expenditure on LTC services in 38 European OECD countries was estimated at 2.1% of GDP, equating to $860

**Data availability statement:** The dataset used in this study has been uploaded to a public database. The dataset is named "OFS" and can be accessed at the following URL: https://osf.io/. The DOI of the dataset is 10.17605/OSF.IO/BN32X.

**Funding:** Project of the National Social Science Foundation of China [grant number 22CJY040];Project of the Department of Educational Science Planning of the Inner Mongolia Autonomous Region [grant number NGJGH2023346]. The funders had no role in study design, data collection and analysis, decision to publish, or preparation of the manuscript.

**Competing interests:** The authors have declared that no competing interests exist.

per capita [7]. In China, the accessibility of formal LTC services is limited, with 80% of LTC services and more than 50% of LTC costs provided and paid for by family members [8]. The substantial costs of LTC can place a significant financial burden on families [6]. Additionally, the interplay between disability and illness often prolongs older people's illnesses, leading to continuous LTC expenditures. This situation is further compounded by the reduction in family labor supply time due to informal caregiving responsibilities, which decreases the income of family caregivers [9]. Consequently, some elderly families risk falling into poverty due to illness [10]. Numerous studies have shown that informal caregiving responsibilities have a substantial negative impact on labor supply and life satisfaction among family members [11,12]. Preventing the risk associated with LTC is a key factor contributing to the high savings rate among elderly households in China [13]. Therefore, given the influence of China's "filial piety" culture and the trend toward smaller family sizes, long-term care insurance (LTCI), known as the "sixth insurance" of social security, is an effective means to reduce LTC expenses for elderly families, free up labor supply time for family caregivers, and increase household consumption capacity. LTCI can significantly affect household expenditures, quality of life, and wealth accumulation among the elderly [14]. However, despite its theoretical benefits, empirical evidence on LTCI's impact on elderly household expenditures, its mechanisms, and its pathways remains underexplored.

LTCI, a form of social health insurance, provides cash compensation or reimbursement for individuals who are unable to perform daily activities due to age, illness, or disability and who require various basic care and treatment [15]. Unlike medical care, which primarily addresses acute health needs, long-term care focuses on supporting the daily living needs of individuals with chronic illnesses or disabilities. Long-term care often integrates certain aspects of medical care to provide comprehensive and continuous support for the elderly, particularly those with disabilities. The primary objective of LTCI is to assist disabled elderly individuals in accessing comprehensive and multi-level healthcare services, as well as financial support [16]. It aims to bridge the gap between acute medical services and the long-term care required for chronic conditions, ensuring continuity and holistic protection for the elderly population [17]. This helps diversify the risk of disability, reduce the pressure of informal care at home, improve the quality of life in later years, and promote social equity [18].

In the 1970s, several European and American countries began developing and establishing LTCI systems. Drawing from these developed countries' experiences and considering its own national conditions, China launched its first batch of LTCI pilots in 15 cities in 2016 (as shown in Table 1). The second batch was expanded to another 14 cities in 2020 [19]. By the end of 2022, there were 49 LTCI pilot cities in China, covering 169 million people, with a cumulative total of 1.95 million people receiving benefits [20]. The aim is to provide comprehensive medical care services and financial support for the disabled elderly. China's LTCI pilot program has three distinctive features. First, the LTCI program is closely integrated with social health insurance [17]. The LTCI fund primarily receives and disburses income and expenses through the Social Health Insurance Fund (SHIF). Although the LTCI program's financing is like that of social health insurance—shared by the government, employees, and individuals—the contribution levels from employees and individuals are too low. Consequently, the LTCI fund's income and expenses are heavily dependent on the SHIF. The LTCI program fully covers participants of the Urban Employees' Basic Medical Insurance (UEBMI), and in some pilot cities, it also extends to participants of the Urban Residents' Basic Medical Insurance (URBMI) and the Urban and Rural Residents' Basic Medical Insurance (URRBMI). Second, the assessment criteria for the LTCI program are tailored to local conditions [21]. The program primarily covers individuals with severe disabilities, but some pilot cities have expanded coverage to include those with moderate disabilities or dementia. The instruments

**Table 1. Timing of implementation, coverage and funding of the first LTCI pilot cities.**

| Pilot Cities | Policy Implementation Time | Insured Person | Sources of Funds |
|---|---|---|---|
| Qingdao[a] | July 2012 | UEBMI&URRBMI | SHIF&UC&IC |
| Jingmen | Nov.2016 | UEBMI&URRBMI | SHIF&GS&IC |
| Shanghai | Jan.2017 | UEBMI&URRBMI | SHIF |
| Suzhou | Jun.2017 | UEBMI&URRBMI | SHIF&GS&IC |
| Shangrao | Nov.2016 | UEBMI | SHIF&GS&IC&UC |
| Chengde | Nov.2016 | UEBMI | SHIF&GS&IC |
| Anqing | Jan.2017 | UEBMI | SHIF&IC |
| Chengdu | July 2017 | UEBMI | SHIF&GS&IC&UC |
| Guangzhou | Aug.2017 | UEBMI | SHIF |
| Qiqihaer | Oct.2017 | UEBMI | SHIF&IC |
| Ningbo | Dec.2017 | UEBMI | SHIF |
| Chongqing | Dec.2017 | UEBMI | SHIF&IC |
| Shihezi[a] | Jan.2017 | UEBMI&URBMI | SHIF&GS&IC |
| Nantong[a] | Jan.2016 | UEBMI&URBMI | SHIF&GS&IC |
| Changchun[a] | May.2015 | UEBMI&URBMI | SHIF&UC&IC |

**Notes:** Data collected from the official websites of provincial and municipal governments. [a] representative did not include this city in this study.

Abbreviations: UEBMI, Urban Employee Basic Medical Insurance; URBMI, Urban Resident Basic Medical Insurance; URRBMI, Urban and Rural Resident Basic health insurance; SHIF, Social Health Insurance Fund; UC, Unit Contributions. IC, Individual Contributions; GS, Government Subsidies.

used to assess disability and the ability to perform activities of daily living (ADL) also vary among pilot cities [22]. Some cities use the Barthel Scale, while others employ self-developed comprehensive assessment tools, leading to a lack of uniformity in disability assessment criteria [23]. Third, LTCI programs in a few pilot cities provide cash subsidies and reimbursements for institutional and nursing home care [24]. They also offer cash incentives for informal caregivers providing home care, aiming to motivate family caregivers and ensure the efficient use of LTC resources.

China's LTCI system has garnered extensive attention from scholars worldwide since its inception [16]. Relevant studies have examined the balance of payments and financial efficiency of LTCI, analyzing the strengths and weaknesses of its system design, operational model, and risk management. However, prior research has predominantly focused on financial sustainability at the macro level, often overlooking service quality and individual experiences at the micro level. Additionally, many studies have concentrated on policy guidelines from specific pilot cities, thus neglecting the broader applicability of China's LTCI pilot initiatives. Furthermore, some studies have assessed the effects of LTCI policies without including populations outside the pilot cities as control groups or without considering key influencing factors in their control variables. This has led to significant variability in findings, often due to omitted variable bias. For example, Deng(2022)found that LTCI was not effective in reducing the financial burdens of families with older adults who have disabilities [25]. In contrast, Feng(2020)found that LTCI significantly alleviated the financial and healthcare burdens of these families [26]. Based on this, this study utilizes the latest 2020 data from the China Health and Retirement Longitudinal Study (CHARLS) and incorporates comprehensive variables, including individual characteristics, family status, and health demands. The aim is to scientifically and quantitatively explore the impact of the Chinese LTCI pilot program on elderly household expenditures and its structural characteristics. Furthermore, the study analyzes the mechanism of LTCI's influence on household expenditures from the perspectives of self-assessed health and economic intergenerational support. The findings aim to provide

feasible policy recommendations to enhance elderly household expenditures and promote the sustainable development of China's LTCI. Additionally, this study aims to contribute to research trends in the field of LTCI.

## 2. Long-term care insurance and household expenditures for the elderly

As a crucial indicator of the economic situation and quality of life of elderly households, as well as a basis for assessing the effects of the LTCI policy, LTCI plays a significant role. Through its risk diversification mechanism, LTCI mitigates health risk shocks within the insurance pool, thereby altering the behavior and structure of elderly household expenditures. On one hand, the most direct impact of LTCI on these expenditures is its ability to significantly reduce household medical costs, which in turn positively influences other household expenditures. According to Angrisani's study, the out-of-pocket ratio for purchased services in elderly households with LTCI entitlement is substantially lower than in those without it when long-term care is needed [27]. Additionally, households with LTCI can further reduce healthcare expenditures through the reimbursement of caregiving costs. Moreover, LTCI can indirectly influence elderly household expenditures by enhancing the predictability of these expenditures and improving the temporal stability of household labor supply [28]. Specifically, when family members know that care services are covered by LTCI reimbursement, they exhibit greater confidence in their financial planning and budgeting for household expenditures, allocating a larger proportion to non-medical expenses. Additionally, formal care services provided by LTCI can replace informal family care, freeing up time for family members and increasing the labor income of family caregivers. There is a significant positive relationship between disposable family income and household expenditures, meaning higher income leads to higher expenditures, a trend especially pronounced in low-income families [29]. Based on this, the study proposes the following hypotheses grounded in existing research and various types of elderly household expenditures:

Hypothetical 1a:LTCI contributes to lowering medical health care expenditures for elderly households;

Hypothetical 1b:LTCI contributes positively to boosting non-medical health care expenditures of elderly households;

Hypothetical 1c:LTCI contributes positively to raising total household expenditures of older adults.

According to the Precautionary Saving Theory, individuals tend to increase precautionary savings and reduce consumption expenditures to build an economic buffer against potential health risks and related uncertainties in the future [30]. Self-assessed health, an individual's subjective judgment and evaluation of their own physical and mental health, is considered a reliable and effective indicator of overall health, directly related to quality of life and well-being. Leng(2024) found that LTCI positively impacts the self-assessed health of the elderly [31]. This improvement effectively reduces the uncertainty of future health risks and household expenditures. Therefore, the effect of LTCI on the household expenditures of the elderly is likely mediated by the reduction of precautionary savings through improved self-assessed health.

Economic intergenerational support primarily refers to the economic exchanges between parent and child generations. In this paper, it specifically denotes the unilateral financial assistance provided by children to their parents. In China, children bear the main responsibility

for parental care and are often the primary providers of financial resources for their parents. From the perspective of Altruism Theory, children prioritize their parents' well-being and health alongside their own interests [32]. The implementation of the LTCI system alleviates the economic burden of long-term care, reducing both the informal care and financial pressures on children [33]. This shared responsibility allows children to participate more in the labor market, increasing their labor income and thereby enhancing economic intergenerational support. Consequently, this promotes higher household expenditures for the elderly. Based on the above analysis, the theoretical framework of this paper is illustrated in Fig 1, and the following research hypotheses are proposed:

Hypothetical 2a: Self-rated health mediates the relationship between LTCI and household expenditures among older adults;

Hypothetical 2b: Economic intergenerational support mediates the relationship between LTCI and household expenditures among older adults.

## 3. Data and methods

### 3.1. China Health and Retirement Longitudinal Study

High-quality micro data from the China Health and Retirement Longitudinal Study (CHARLS) for the years 2015, 2018, and 2020 were selected for this empirical study. CHARLS is a long-term tracking survey program jointly conducted by the National Academy of Social Sciences at Peking University and the Peking University Youth League Committee. It aims to explore and study the health status, economic status, social participation, retirement life, and pension status of China's middle-aged and elderly population [34]. The CHARLS database covers 19,000 individuals across 28 provinces, autonomous regions, and municipalities in China, despite limitations due to survey resources, data quality, and feasibility considerations. The research team conducted the baseline survey in 2011 and followed up with subsequent surveys in 2013, 2015, 2018, and 2020, ensuring a high-quality and representative sample. CHARLS has been approved by the Biomedical Ethics Review Committee of Peking University and assigned the ethical approval number IRB00001052–11015. All participants were required to provide written informed consent.

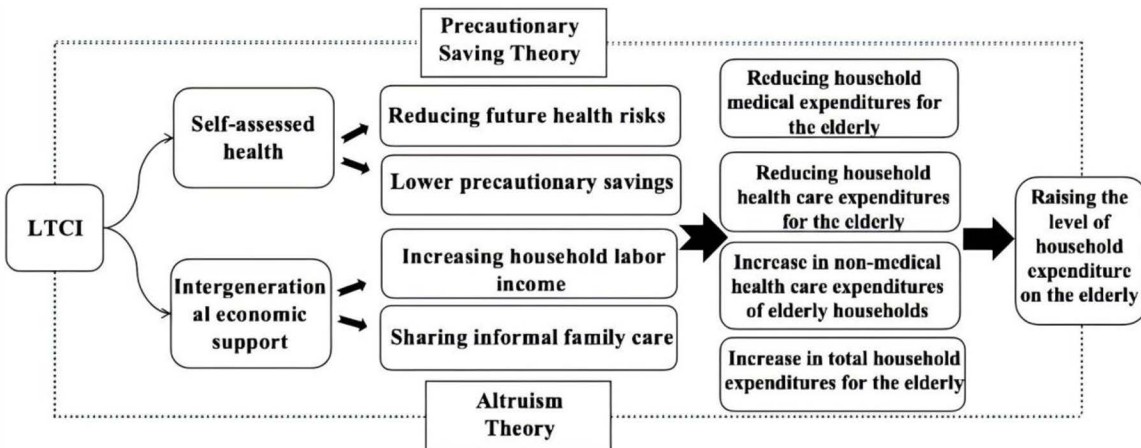

**Fig 1. Theoretical framework of the LTCI influencing household expenditures of older adults.**

The three-period sample data used in this study covers the period before and after the implementation of China's LTCI policy, focusing on individuals aged 60 and above. Eleven cities from the first pilot cities of LTCI were selected as the treatment group, while the remaining non-pilot provinces and cities served as the control group. There are two reasons for selecting only 11 pilot cities as the treatment group: first, Jilin Province and Shandong Province, key areas for the LTCI pilot work, initiated their pilots in some cities before the Ministry of Human Resources and Social Security introduced the LTCI system in 2016, making them unsuitable for the study timeline. To ensure data quality and feasibility, Jilin Province, Shandong Province, Changchun City, and Qingdao City were excluded from the sample. Second, the CHARLS project research team did not collect information in Nantong City and Shihezi City, resulting in no relevant sample data for these cities, necessitating their exclusion. After removing invalid samples and missing values, a balanced panel data set was obtained, comprising 9,029 samples, 689 in the treat group and 8340 in the control group.

## 3.2. Variable design

The dependent variables in this study are household expenditures of the elderly, measured using four variables: household medical expenditures (HME), household health care expenditures (HHCE), household non-medical health care expenditures (HNMHCE), and total household expenditures (THE). HME refer to expenditures on medical services and products, encompassing both direct costs (e.g., hospital fees, medication) and indirect costs (e.g., transportation for medical visits). This is measured using the indicator titled "Your family's expenditures on medical consumption in the past year" from the CHARLS database. HHCE include expenditures on maintaining or improving health but are not directly related to medical treatments. This category covers costs for fitness and exercise, exercise products and equipment, and health care products, as indicated by the same title from the CHARLS database. HNMHCE are defined as expenditures on all other categories except for health-related items, such as general household consumption, as measured by the CHARLS question, "How much did your family spend in the past year on all categories of consumption except medical health care?" It is important to note that HME and HHCE are distinct and mutually exclusive categories, as medical expenditures are strictly limited to treatment-related costs, whereas health care expenditures include preventive and wellness-related spending. HNMHCE, by contrast, includes all other expenditures unrelated to medical or health care. This includes daily living expenditures, education expenditures, and social contributions.

Additionally, the question "On average, how much did your family spend in a month?" from the database was multiplied by 12 to measure the total annual expenditures of elderly families. Together, these categories provide a comprehensive framework to analyze household expenditures among elderly families. To ensure the accuracy of the analysis, we examined the level of missingness for the expenditure variables. The results showed that the percentage of missing values was as follows: HME had a missing rate of 3.5%, HHCE 4.2%, HNMHCE 2.8%, and total household expenditures (THE) 3.9%. To handle missingness, we employed listwise deletion, a commonly used method in empirical studies, given that the overall level of missingness was below 5%. This method involves excluding observations with missing values for any variable used in the analysis [35]. The rationale for choosing this approach is its simplicity and minimal risk of introducing biases when the level of missingness is low. As a result, the final sample size for our analysis was reduced from 9417 to 9029 households.

In empirical research, taking the logarithms of expenditure or income variables can make the data more consistent with the assumptions of the regression model, improve the explanatory power and predictive accuracy of the model, and help reduce numerical bias and the

impact on regression results [36,37]. Therefore, in this study, all variables involving expenditure or income are logarithmized [38].

The key variable of this study is participation in LTCI (DID). This variable is constructed from the interaction terms of the LTCI pilot city dummy variable (Treat) and the LTCI pilot time dummy variable (Time), such that DID = Treat * Time. The LTCI pilot city dummy variable is based on the list of pilot cities in Table 1. For example, individuals must reside in a pilot city and be covered by their local government regulations (e.g., Chengdu city requires enrollment in the UEBMI, while Shanghai city requires participation in either the UEBMI or the URRBMI). The LTCI pilot time dummy variable uses the release of China's LTCI policy in 2016 as the reference point. Years 2018 and 2020, being after the policy release, are assigned a value of 1, while the year 2015, being before the policy release, is assigned a value of 0.

To investigate whether demographic characteristics, family relationships, and social needs impact household expenditures of the elderly, this study divides the control variables into three categories: individual characteristics, family, and social support. The individual characteristics category includes seven variables: Age, Gender, Education, Marital status, Living arrangement, Smoking status, and Clinic visits. The family category includes one variable: Annual family income. The social support category includes one variable: Pension.

Combining the theoretical analysis and research hypotheses, this study selects Self-assessed health and Intergenerational economic support as mediating variables to explore the mechanism of LTCI on the household expenditures of the elderly. Specific variable definitions and assignments are detailed in Table 2.

## 3.3. Empirical strategies

The LTCI policy pilot provides a quasi-natural experimental condition to test the impact of the LTCI system on household expenditures of the elderly. To address the lack of objectivity and randomness in evaluating LTCI's effect and to reduce the impact of endogeneity and self-selection bias, this study employs Propensity Score Matching (PSM) to ensure sample balance and achieve "counterfactual" estimation. Specifically, PSM-DID is used to empirically test the impact of LTCI on household expenditures of the elderly. Following the methodologies of Yang(2019), the detailed model is presented below [39]:

$$Y_{it}^{psm} = \beta_0 + \beta_1 Treat_i + \beta_2 Time_t + \beta_3 Treat_i \times Time_t + \beta_4 Control_{it} + \varepsilon_{it} \qquad (1)$$

In formula (1), $i$ denotes an elderly household and $t$ denotes the time period, $y_{it}$ represents the expenditure of elderly household $i$ in period $t$. The variable $Treat_i$ is 1 for the treatment group and 0 for the control group. The variable $Time_t$ is 0 before the policy implementation and 1 after. The interaction term $Treat_i \times Time_t$ captures the policy effect in the DID framework; $Control_{it}$ represents control variables, and $\varepsilon_{it}$ is the random error term.

$$Y_{it} = \beta_0 + \beta_1 Treat_i + \beta_2 Time_t + \beta_3 Treat_i \times Time_t + \beta_4 Control_{it} + \varepsilon_{it} \qquad (2)$$

$$M_{it} = \theta_0 + \theta_1 Treat_i + \theta_2 Time_t + \theta_3 Treat_i \times Time_t + \theta_4 Control_{it} + \varepsilon_{it} \qquad (3)$$

$$Y_{it+1} = \alpha_0 + \alpha_1 Treat_i + \alpha_2 Time_t + \alpha_3 Treat_i \times Time_t + -_4 M_{it} + \alpha_5 Control_{it} + \alpha_6 Y_{it} + \varepsilon_{it} \qquad (4)$$

Equations (2) through (4) are the mediation effect test models [40]. In Equation (4), we employed the system Generalized Method of Moments (GMM) estimation method to address

**Table 2. Descriptive statistics and assignment of variables.**

| Variables | Definition and assignment | Full (N=9029) | | Treated (N=689) | | Control (N=8340) | |
|---|---|---|---|---|---|---|---|
| | | Mean | S.D. | Mean | S.D. | Mean | S.D. |
| **Key variable** | | | | | | | |
| DID | Participation in LTCI = 1; No participation = 0 | 0.076 | 0.265 | 1.000 | 0.000 | 0.000 | 0.000 |
| **Dependent variables** | | | | | | | |
| HME | Log(HME+1) | 6.548 | 3.160 | 6.437 | 3.284 | 6.557 | 3.149 |
| HHCE | Log(HHCE+1) | 5.956 | 3.409 | 6.432 | 3.229 | 5.916 | 3.421 |
| HNMHCE | Log(HNMHCE+1) | 2.999 | 3.046 | 3.088 | 2.972 | 2.992 | 3.052 |
| THE | Log(THE+1) | 8.955 | 1.362 | 9.521 | 1.101 | 8.909 | 1.371 |
| **Control variables** | | | | | | | |
| Gender | Male = 1; Female = 0 | 0.589 | 0.491 | 0.640 | 0.480 | 0.585 | 0.492 |
| Age | 60 years old and above | 69.717 | 6.119 | 70.354 | 5.544 | 69.664 | 6.162 |
| Education | Elementary school = 1; Middle school = 2; High school and above = 3 | 1.321 | 0.616 | 1.265 | 0.581 | 1.326 | 0.618 |
| Marital | Married = 1; Unmarried = 0 | 0.803 | 0.397 | 0.834 | 0.371 | 0.800 | 0.399 |
| Live | Urban = 1; Rural = 0 | 0.249 | 0.432 | 0.283 | 0.45 | 0.247 | 0.431 |
| Smoke | Smoking = 1; Non-smoking = 0 | 0.328 | 0.469 | 0.37 | 0.483 | 0.325 | 0.468 |
| Clinic | Yes = 1; No = 0 | 0.195 | 0.396 | 0.174 | 0.379 | 0.197 | 0.398 |
| Pension | Participation = 1; Non-participation = 0 | 0.835 | 0.370 | 0.865 | 0.341 | 0.832 | 0.373 |
| Annual family income | Log(Annual family income+1) | 8.881 | 2.221 | 9.544 | 1.807 | 8.826 | 2.243 |
| **Mediating variables** | | | | | | | |
| Self-assessed health | 1 to 5, with higher scores indicating better health | 2.931 | 0.963 | 3.004 | 0.909 | 2.925 | 0.967 |
| Intergenerational economic support | Log(Intergenerational economic support+1) | 6.677 | 3.215 | 5.792 | 3.675 | 6.750 | 3.163 |

**Note:** We restrict the sample to households with at least one member aged 60 and above.

the potential endogeneity issue introduced by the inclusion of $Y_{it}$, the lagged dependent variable. The system GMM approach effectively mitigates these concerns by constructing instruments based on the lagged levels and first differences of $Y_{it}$. Specifically, it uses lagged levels as instruments for the first-difference equation and first differences as instruments for the level equation. This dual framework leverages the orthogonality conditions between the instruments and the error term, ensuring unbiased and consistent parameter estimation. According to Hypotheses 2a and 2b, LTCI likely affects household expenditures of the elderly through the mediating roles of self-assessed health and intergenerational economic support. To test whether self-assessed health and intergenerational economic support mediate the relationship between the LTCI system and household expenditures of the elderly, we employ established mediation effect analysis methods.

## 4. Results

### 4.1. DID results

The regression results of the effect of LTCI on household expenditures of the elderly are presented in Table 3. According to columns (1) and (2), the coefficients of the impact of LTCI on household medical expenditures (HME) and household health care expenditures (HHCE) are -0.453 and -0.457, respectively, both significant at the 5% level. This indicates that LTCI significantly reduces HME and HHCE, thus supporting Hypothesis 1a. Column (3) shows the relationship between LTCI and household non-medical health care expenditures (HNMHCE), with a coefficient of 0.646, significant at the 1% level. This indicates that LTCI significantly

**Table 3. Benchmark regression results for the impact of the LTCI on household expenditures of the elderly.**

| Variables | (1) | (2) | (3) | (4) |
|---|---|---|---|---|
| | HME | HHCE | HNMHCE | THE |
| DID | -0.453** | -0.457** | 0.646*** | 0.241*** |
| | (0.229) | (0.229) | (0.209) | (0.078) |
| _cons | 3.986*** | 5.382*** | 2.339*** | 7.560*** |
| | (0.444) | (0.444) | (0.404) | (0.151) |
| Control variables | Y | Y | Y | Y |
| Individual FE | Y | Y | Y | Y |
| Year FE | Y | Y | Y | Y |
| R-squared | 0.063 | 0.195 | 0.071 | 0.416 |
| Observations | 9029 | 9029 | 9029 | 9029 |

**Note:** The significance levels of 1%, 5%, and 10% are denoted by

***,

**, and

*, respectively. Standard errors in parentheses.

increases HNMHCE, validating Hypothesis 1b. Finally, LTCI positively affects total household expenditures (THE) with a coefficient of 0.241, significant at the 1% level, supporting Hypothesis 1c. Specifically, LTCI helps to significantly reduce HME and HHCE while significantly increasing HNMHCE and THE. This shift suggests that the implementation of the LTCI system allows elderly households to allocate more economic resources to HNMHCE, indicating an increase in their disposable expenditure capacity and thus raising their overall household expenditure level. In addition, since the LTCI policy positively impacts the total expenditure of elderly households, its role in raising HNMHCE is considered more significant than its role in reducing HME and HHCE.

## 4.2. Robustness test results

**4.2.1. PSM.** Propensity Score Matching (PSM) was employed to compare the means before and after matching for the Treated and Control groups. This was done to evaluate the effectiveness of the matching process in achieving covariate balance, detect the matching effect, assess the sample quality, and examine the endogeneity and balance of the treatment effect. The specific test results are presented in Table 4.

The findings indicate that the absolute value of the standard deviation for each variable after matching is less than 10%, suggesting minimal differences in the mean values of the variables. Additionally, the P-values of the variables after matching are not significant, reflecting high-quality matching. For instance, the percentage change in the variable "Annual family income" decreased from 35.3% to -0.6% after matching, and its P-value shifted from significant (0.000) to non-significant (0.899). This indicates that the distribution of this variable is more balanced between the treatment and control groups post-matching, thereby reducing bias in estimating the treatment effect. Overall, the sample passes the balance test, ensuring the reliability of the matching results [41]. This process effectively mitigates the impact of selection bias and endogeneity problems caused by the non-random distribution of treatment on the benchmark regression results.

To further assess the matching effect and quality of the samples, the kernel density function of propensity scores was analyzed before and after matching. Fig 2 illustrates that, before matching, the kernel density distributions of the treatment and control groups differ significantly. However, after matching, the kernel density plot shows that the propensity score values

**Table 4. Results of the balance test after PSM.**

| Variable | Sample Matching | Matched Mean | | % bias | % Reduct (bias) | t-Test | |
|---|---|---|---|---|---|---|---|
| | | Treated | Control | | | T | *p*>t |
| Gender | Unmatched | 0.640 | 0.585 | 11.2 | 76.0 | 2.79 | 0.005 |
| | Matched | 0.640 | 0.653 | -2.7 | | -0.51 | 0.612 |
| Age | Unmatched | 70.354 | 69.665 | 11.8 | 66.1 | 2.84 | 0.004 |
| | Matched | 70.354 | 70.588 | -4 | | -0.74 | 0.460 |
| Education | Unmatched | 1.265 | 1.326 | -10.1 | 71.2 | -2.47 | 0.013 |
| | Matched | 1.265 | 1.283 | -2.9 | | -0.56 | 0.578 |
| Marital | Unmatched | 0.834 | 0.8 | 8.7 | 95.7 | 2.13 | 0.033 |
| | Matched | 0.834 | 0.833 | 0.4 | | 0.07 | 0.942 |
| Live | Unmatched | 0.283 | 0.247 | 8.1 | 91.9 | 2.09 | 0.036 |
| | Matched | 0.283 | 0.28 | 0.7 | | 0.12 | 0.905 |
| Smoke | Unmatched | 0.37 | 0.325 | 9.4 | 83.8 | 2.41 | 0.016 |
| | Matched | 0.37 | 0.377 | -1.5 | | -0.28 | 0.781 |
| Clinic | Unmatched | 0.174 | 0.197 | -6 | 87.6 | -1.48 | 0.138 |
| | Matched | 0.174 | 0.177 | -0.7 | | -0.14 | 0.887 |
| Annual family income | Unmatched | 9.544 | 8.826 | 35.3 | 98.4 | 8.19 | 0.000 |
| | Matched | 9.544 | 9.556 | -0.6 | | -0.13 | 0.899 |
| Pension | Unmatched | 0.865 | 0.832 | 9.0 | 41.1 | 2.18 | 0.029 |
| | Matched | 0.865 | 0.883 | -5.3 | | -1.06 | 0.291 |

**Note:** The table shows the results of the nearest neighbor matching (1:1) test.

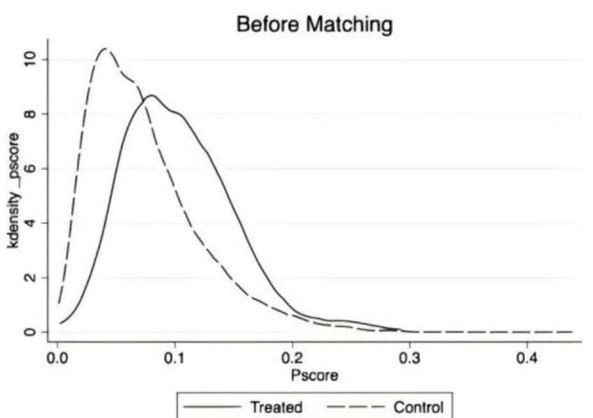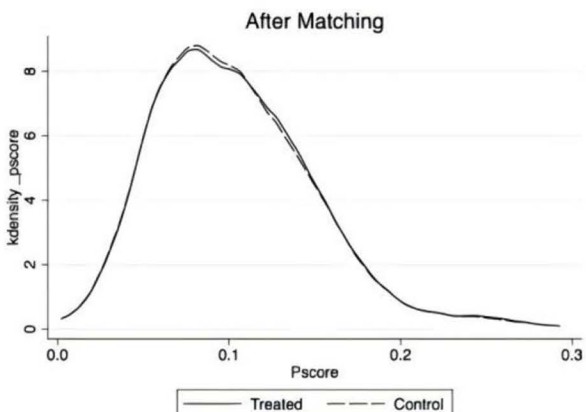

**Fig 2. Propensity score values before and after matching.**

of the treatment and control groups become more consistent, with significant proximity and coverage. This indicates an improved matching effect and suggests that the sample passes the equilibrium test (the test of common support). Consequently, this process effectively mitigates issues of selection bias and endogeneity.

**4.2.2. Individual Placebo Test.** In this study, 500 replicated randomized experiments were conducted to test the model, and the estimated coefficients were plotted, as shown in Fig 3. The results indicate that the kernel density estimation curves for the four different expenditure scenarios are tightly clustered around zero, suggesting that the corresponding

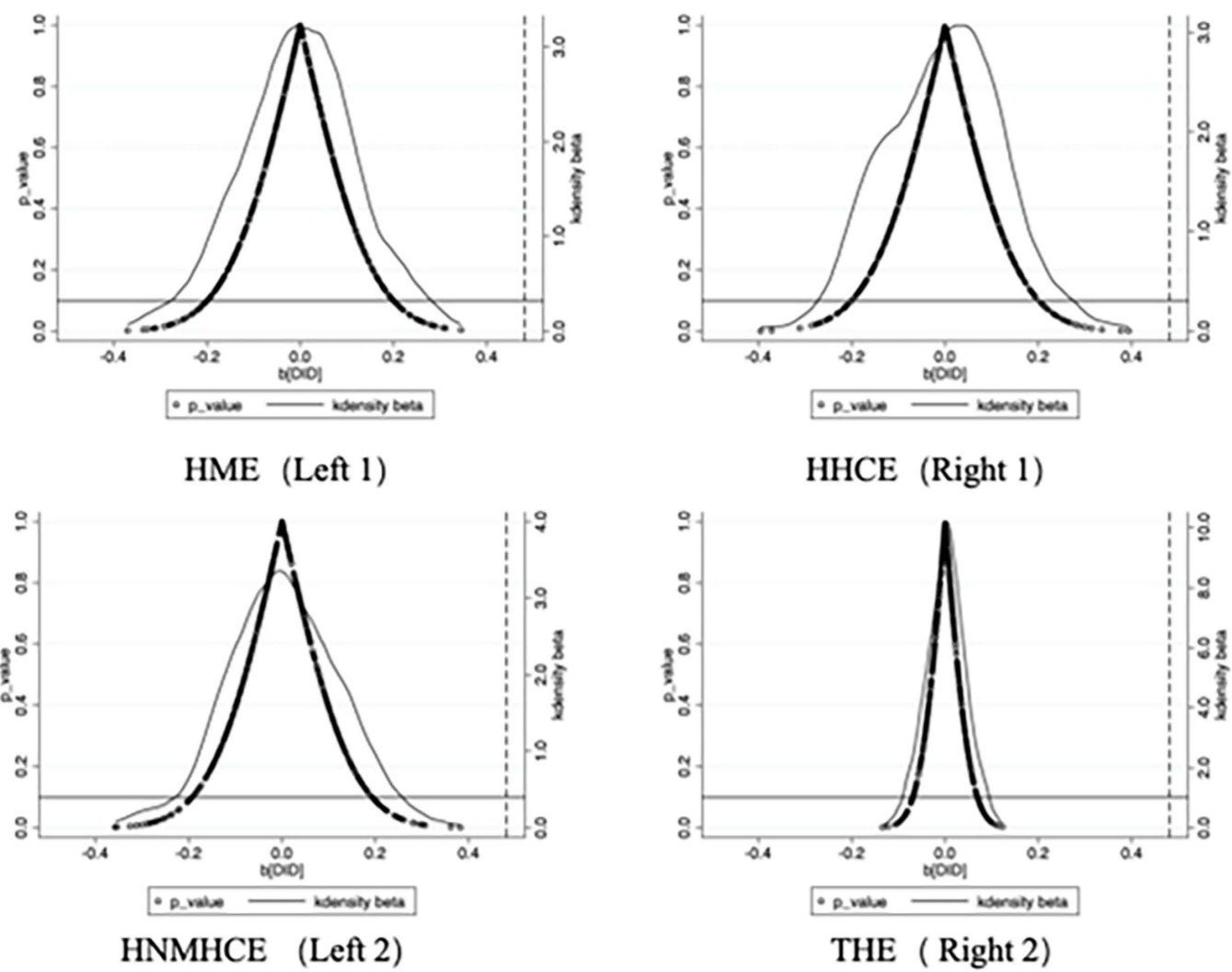

HME (Left 1) HHCE (Right 1)

HNMHCE (Left 2) THE (Right 2)

**Fig 3. Kernel density diagram of interaction term coefficient distribution.**

effect estimates are mostly close to a no-effect state [42]. Furthermore, the estimates resemble the normal distribution curve, with the mean of the estimated coefficients distributed near zero. The corresponding P-values are not concentrated below the traditional significance level, indicating that the effect under the placebo scenario is not significant. This implies that the effect of LTCI on HME, HHCE, HNMHCE, and THE is not a chance event, further confirming the high reliability and validity of the study's findings and supporting the robustness of the DID (Baseline regression) results.

**4.2.3. Consideration of Price indices and replacement of Dependent variables.** Elderly household expenditure is influenced by the price consumption index and inflation. To eliminate this objective effect, this study summarizes the real growth index of residents' various consumption expenditures, using 2015 as the base year. The indices of residents' mean of production, consumption levels, and healthcare expenditures for the current year are normalized to 100, and the corresponding indices for various consumption expenditures in 2020 are deflated to reflect the actual expenditure levels after accounting for inflation. The results, presented in Table 5, indicate that the correlation and significance of the impact

**Table 5. DID regression results after considering price indices.**

| Variables | (1) | (2) | (3) | (4) |
|---|---|---|---|---|
| | HME | HHCE | HNMHCE | THE |
| DID | -0.379** | -0.382** | 0.540*** | 0.201*** |
| | (0.192) | (0.192) | (0.174) | (0.065) |
| _cons | 3.333*** | 4.500*** | 1.956*** | 6.321*** |
| | (0.371) | (0.371) | (0.338) | (0.126) |
| Control variables | Y | Y | Y | Y |
| Individual FE | Y | Y | Y | Y |
| Year FE | Y | Y | Y | Y |
| R-squared | 0.064 | 0.195 | 0.072 | 0.417 |
| Observations | 9029 | 9029 | 9029 | 9029 |

**Note:** Price indices from China Bureau of Statistics. The significance levels of 1%, 5%, and 10% are denoted by

***,

**, and

*, respectively. Standard errors in parentheses.

coefficients align with the results of the DID regression. This supports the robustness of the effect of LTCI on the expenditure levels of elderly households.

To address the remaining endogeneity problem and its correlation bias, this study employs a robustness test by replacing the dependent variables. The various household expenditure variables used previously were replaced with per capita household medical expenditures (PCHME), per capita household health care expenditure (PCHHCE), per capita household non-health care expenditure (PCHNMHCE), and per capita total household expenditure (PCTHE). The results, presented in Table 6, show that the regression outcomes with the new dependent variables align with the baseline regression results in Table 2. This indicates that the empirical findings of this study are highly reliable and valid.

## 4.3. Heterogeneity test results

The urban-rural dichotomy serves as a prominent context for LTCI piloting in China. This study examines the urban-rural heterogeneity of LTCI policy effects by analyzing differences in household expenditures for the elderly. The regression results, presented in Table 7, reveal a clear disparity in the effects of LTCI on total health expenditures (THE) between urban and rural areas. Specifically, LTCI does not significantly impact urban THE, as shown in Column (4). However, LTCI significantly contributes to rural THE at the 1% level, demonstrating a pronounced promoting effect on rural households' expenditures. This disparity highlights the differential roles of LTCI in urban and rural settings. Urban older adults typically have access to more extensive and well-developed healthcare resources and social care services. Furthermore, urban residents often possess greater savings and pensions, enabling them to diversify health risks and reduce reliance on LTCI. Consequently, LTCI has a limited impact on urban THE. In contrast, rural households, with less access to comprehensive social care and fewer financial resources, are more likely to rely on LTCI as a critical health protection mechanism.

Table 8. analyzes the impact of LTCI on household expenditures of older adults with different education levels, dividing the study population into three groups. The results indicate that LTCI has a greater effect on increasing household expenditures for the elderly with lower education levels. This effect likely arises from the correlation between income and expenditure, where education level significantly and positively affects income. Lower education levels

**Table 6. Replacing the dependent variables.**

| Variables | (1) | (2) | (3) | (4) |
|---|---|---|---|---|
|  | PCHME | PCHHCE | PCHNMHCE | PCTHE |
| DID | -0.363* | -0.385* | 0.611*** | 0.275*** |
|  | (0.214) | (0.214) | (0.201) | (0.079) |
| _cons | 3.091*** | 4.331*** | 1.612*** | 6.431*** |
|  | (0.414) | (0.415) | (0.389) | (0.153) |
| Control variables | Y | Y | Y | Y |
| Individual FE | Y | Y | Y | Y |
| Year FE | Y | Y | Y | Y |
| R-squared | 0.061 | 0.223 | 0.064 | 0.446 |
| Observations | 9029 | 9029 | 9029 | 9029 |

**Note:** The significance levels of 1%, 5%, and 10% are denoted by

***,

**, and

*, respectively. Standard errors in parentheses.

**Table 7. Live heterogeneity test.**

| Variables | Observations | (1) | (2) | (3) | (4) |
|---|---|---|---|---|---|
|  |  | HME | HHCE | HNMHCE | THE |
| Urban | 2256 | -0.982** | -0.824* | 0.883*** | 0.082 |
|  |  | (0.473) | (0.432) | (0.395) | (0.133) |
| R-squared |  | 0.072 | 0.202 | 0.069 | 0.51 |
| Rural | 6773 | -0.300* | -0.359* | 0.591*** | 0.292*** |
|  |  | (0.263) | (0.27) | (0.245) | (0.093) |
| R-squared |  | 0.053 | 0.154 | 0.060 | 0.332 |

**Note:** The significance levels of 1%, 5%, and 10% are denoted by

***,

**, and

*, respectively. Standard errors in parentheses. All regressions incorporated Control variables, Individual FE and Year FE.

are usually associated with lower income and fewer economic resources, so LTCI more significantly boosts household expenditure for the elderly with lower education levels [43].

## 5. Mechanism test

### 5.1. Analysis of mediating effect of Self-assessed health

Table 9 presents the results of the mediating effect of self-assessed health. First, column (1) shows that the implementation of the LTCI system significantly enhances the self-assessed health of the elderly. Second, columns (2) to (4) indicate that self-assessed health positively influences household expenditures of the elderly. It has a significant negative correlation with HME and HHCE, and a significant positive effect on HNMHCE and THE. Specifically, improved self-assessed health among older adults significantly reduces HME and HHCE, while increasing HNMHCE and THE. Finally, when self-assessed health is included as a mediating variable, the regression results show that the effect of LTCI on HME loses its significance but remains significant for the other three types of household expenditures. This suggests

**Table 8. Education heterogeneity test.**

| Variables | Observations | (1) | (2) | (3) | (4) |
|---|---|---|---|---|---|
| | | HME | HHCE | HNMHCE | THE |
| Elementary school | 6586 | -0.373* | -0.351* | 0.628*** | 0.341*** |
| | | (0.252) | (0.257) | (0.234) | (0.089) |
| R-squared | | 0.057 | 0.168 | 0.059 | 0.368 |
| Middle school | 1444 | -0.674 | -0.821 | 0.759 | 0.016 |
| | | (0.666) | (0.623) | (0.568) | (0.194) |
| R-squared | | 0.066 | 0.170 | 0.055 | 0.487 |
| High school and above | 729 | -0.846 | -0.808 | 0.03 | 0.649* |
| | | (1.159) | (1.063) | (0.969) | (0.336) |
| R-squared | | 0.085 | 0.186 | 0.074 | 0.499 |

**Note:** The significance levels of 1%, 5%, and 10% are denoted by

***,

**, and

*, respectively. Standard errors in parentheses. All regressions incorporated Control variables, Individual FE and Year FE.

**Table 9. Mediating effect of Self-assessed health.**

| | (1) | (2) | (3) | (4) | (5) |
|---|---|---|---|---|---|
| | Self-assessed health | HME | HHCE | HNMHCE | THE |
| DID | 0.156** | -0.350 | -0.472** | 0.659*** | 0.241*** |
| | (0.071) | (0.224) | (0.229) | (0.209) | (0.078) |
| Self-assessed health | | -0.666*** | -0.089*** | 0.079** | 0.064* |
| | | (0.033) | (0.034) | (0.031) | (0.116) |
| Control variables | Y | Y | Y | Y | Y |
| Individual FE | Y | Y | Y | Y | Y |
| Year FE | Y | Y | Y | Y | Y |
| R-squared | 0.046 | 0.102 | 0.196 | 0.072 | 0.416 |

**Note:** According to the results above, the coefficients of the impact of LTCI on all household expenditures of the elderly are significant, which lays the necessary foundation for further in-depth analysis of this paper to see whether self-assessed health and Intergenerational economic support play a mediating role in the relationship. The significance levels of 1%, 5%, and 10% are denoted by

***,

**, and

*, respectively. Standard errors in parentheses.

that self-assessed health fully mediates the relationship between LTCI and HME and partially mediates the relationship between LTCI and the other three expenditures. In other words, self-assessed health mediates the effect of LTCI in promoting household expenditures for the elderly. It indirectly increases other household expenditures by lowering HME and HHCE, thus supporting Hypothesis 2a of this paper.

## 5.2. Analysis of mediating effect of intergenerational economic support

Table 10 presents the results of the mediating effect of intergenerational economic support. The results show that LTCI significantly increases intergenerational economic support for elderly families. This may be because LTCI enables caregivers of elderly family members to

**Table 10. Mediating effect of Intergenerational economic support.**

| | (1) | (2) | (3) | (4) | (5) |
|---|---|---|---|---|---|
| | Intergenerational economic support | HME | HHCE | HNMHCE | THE |
| DID | 0.401* | -0.435* | -0.450** | 0.643*** | 0.245*** |
| | (0.237) | (0.22)) | (0.229) | (0.209) | (0.078) |
| Intergenerational economic support | | 0.046*** | 0.018* | 0.078* | 0.011*** |
| | | (0.01) | (0.01) | (0.092) | (0.03) |
| Control variables | Y | Y | Y | Y | Y |
| Individual FE | Y | Y | Y | Y | Y |
| Year FE | Y | Y | Y | Y | Y |
| R-squared | 0.032 | 0.065 | 0.196 | 0.071 | 0.418 |

**Note:** According to the results above, the coefficients of the impact of LTCI on all household expenditures of the elderly are significant. The significance levels of 1%, 5%, and 10% are denoted by

***,

**, and

*, respectively. Standard errors in parentheses.

reduce informal care duties, thereby gaining more time for labor and improving their income and financial support to their parents. The coefficients for the effect of intergenerational economic support on all expenditures of elderly families are positive and statistically significant at different levels, indicating that intergenerational economic support significantly enhances all types of expenditures for elderly families. Furthermore, LTCI influences the expenditure structure of elderly families through intergenerational economic support, effectively promoting an increase in their expenditure levels. Therefore, intergenerational economic support plays a mediating role in the relationship between LTCI and household expenditures. This confirms Hypothesis 2b of this paper.

## 6. Discussion and conclusion

Enhancing household expenditures for the elderly helps optimize the economic structure and accelerates the quality and upgrading of consumption. Given the growing demand for LTC services among the elderly in China, the rapid expansion of China's LTCI has been inevitable. However, the effects of this insurance scheme on household expenditures and its impact mechanisms have not been fully explored. This study empirically examines the impact of China's LTCI policy pilots on elderly household expenditures using three waves of CHARLS panel data from 2015, 2018, and 2020. We have three key findings.

First, the introduction of LTCI significantly reduces medical and healthcare expenditures for elderly households while significantly boosting non-medical health care and total household expenditures. This evidence suggests that LTCI effectively covers the cost of required LTC services and reduces out-of-pocket payments, consistent with Lu (2020) finding that the LTCI pilot in Qingdao reduced residents' health insurance expenditures and out-of-pocket healthcare costs by 7,918 RMB and 2,324 RMB, respectively [44]. LTCI programs often include health management and preventive care services that help older adults maintain better health, thereby reducing routine healthcare expenditures. More importantly, LTCI alleviates the informal care burden on families, allowing caregivers to focus more on their work and enhance their labor income, which indirectly improves families' disposable spending capacity [45]. Additionally, LTCI has the potential to reduce the need for substantial preventive savings, as it provides financial protection against long-term care costs. While this study does not directly measure the impact of reduced preventive savings on well-being, the evidence

suggests that LTCI can contribute to improved financial stability for households [46]. This, in turn, may allow families to allocate resources to other areas, which could positively influence their quality of life. Overall, the LTCI policy reduces health care expenditures for elderly households by providing comprehensive and professional long-term care services. This minimizes the informal care needs of elderly households, enhances the stability of the labor supply of family members, ensures the continuity of economic resource acquisition, and unleashes the consumption potential of elderly households. As a result, the level of expenditures of elderly households increases, improving the quality of life for both the elderly and their families, these findings align with those of other scholars [47–49].

Secondly, the study results show that LTCI has a greater impact on expenditures in rural areas and among older households with lower levels of education. Specifically, LTCI contributes more significantly to the expenditures of these two groups, indicating that its impact on household expenditures is substantially heterogeneous. In urban areas, abundant medical resources and relatively high-income levels provide the elderly with access to comprehensive LTC services and various means to diversify health risks, making them less dependent on LTCI [50]. In contrast, rural areas face scarce medical resources, limited accessibility to LTC services, and poorer economic conditions, leading elderly households in these areas to rely more on LTCI to alleviate medical and healthcare expenditures [51]. Rural households may rely more on social insurance, including LTCI, to mitigate health risk shocks [52]. Additionally, there are educational differences in the impact of LTCI on elderly households' expenditures. Households with lower levels of education, lacking knowledge in health management and financial planning, and having limited financial resources, are more dependent on LTCI to mitigate medical and healthcare costs. Conversely, better-educated households possess stronger health awareness and financial planning abilities, higher incomes, and a better understanding of insurance and welfare policies, allowing them to seek various welfare resources and reduce their reliance on LTCI [53].

Third, self-assessed health and intergenerational economic support play crucial bridging roles in the impact chain between LTCI and older adults' household expenditures. Specifically, self-assessed health fully mediates the relationship between LTCI and household medical expenditures. By improving the overall health status of older adults, self-assessed health reduces medical and healthcare expenditures while increasing non-medical health care and total household expenditures. LTCI provided services and health management help older adults maintain better health, thereby reducing the need for frequent medical care. Intergenerational economic support also serves as a significant intermediary. LTCI reduces the burden of informal caregiving on family members, allowing them to focus more on work and increase their labor income. This additional income can be used to support household expenses, further easing the financial burden and enhancing overall spending power. In summary, the LTCI pilot has effectively promoted higher household expenditures among the elderly. This paper offers several recommendations based on these findings.

First, expanding LTCI coverage and improving the quality-of-care services are essential. LTCI should be promoted from a pilot program to full coverage across all regions, both urban and rural, ensuring comprehensive access for all residents. Strengthening government financial support and enhancing regulatory frameworks are necessary to ensure the system's sustainability and fairness. Efforts should be made to broaden LTC services, reduce the gap between urban and rural areas in terms of professional LTC services, and promote innovation in service delivery. Improving the quality of caregivers through the establishment of professional standards and continuing education will raise the level of professionalization in LTC services. This approach will collectively address the LTC needs of the elderly and significantly improve their quality of life. Furthermore, need to explore more sustainable and cost-effective

care models in light of LTCI's impact. Specifically, a care model that integrates community and family care could complement LTCI by leveraging existing community resources and familial support systems. This approach may be particularly relevant for rural households, where LTCI significantly alleviates financial burdens and enhances access to care. Reducing reliance on traditional institutional care not only aligns with the preferences of many older adults but also helps optimize resource allocation in regions with limited healthcare infrastructure. Moreover, integrating information technology into LTC services can further enhance the efficiency and personalization of care delivery. For instance, technology can facilitate remote health monitoring, streamline service coordination, and provide tailored interventions, especially for elderly individuals in underserved areas. The benefits of LTCI can be amplified.

Second, the implementation of LTCI should be intensified to unlock the potential for increased household spending by the elderly. The government should increase financial support, optimize the policy framework, and lower the threshold for participation in LTCI. Scientific disability assessment standards and evaluation systems should be developed to ensure that residents in genuine need are not excluded due to unscientific criteria. Enhancing the stability, diversity, and sustainability of LTCI funding sources, rather than relying excessively on the social health insurance fund, is crucial. It is also essential to ensure that elderly families are fully aware of the LTCI policy and its service catalog. Efficient information dissemination and education can increase participation rates and satisfaction. Encouraging pilot cities to expand the scope of the LTCI program and improve the payment structure, considering their unique characteristics and economic strengths, is important. Designing reasonable catalogs and reimbursement contents will provide more beneficiary-friendly protection and reduce the incentive for preventive savings, thus enhancing the economic vitality and consumption capacity of elderly households.

Finally, improving the construction of the LTCI institutional system and formulating differentiated policy measures are imperative. Regional differentiation strategies should be promoted to accommodate the diverse economic and demographic structures across various regions. Additionally, synergies and complementarities between policies should be enhanced. Pilot cities should focus on the LTC service needs of low-income groups, rural areas, and individuals with low education levels, with policies favoring elderly households to increase the relevance and effectiveness of the LTCI system. Establishing service standards, strengthening funding supervision, and enhancing public awareness and participation in the LTCI system are crucial. Policies should be regularly evaluated and optimized through a dynamic adjustment mechanism to ensure the LTCI system remains current. This approach will help clear concerns about household expenditures among the elderly, enabling them to realize and exercise their consumption ability. Ultimately, these measures will improve the well-being and quality of life for elderly family members and promote social harmony.

## Author contributions

**Conceptualization:** Tianshu Zhang, Zeping Hu.

**Data curation:** Kaiyue Zhang.

**Formal analysis:** Zeping Hu, Xinran Li.

**Software:** Kaiyue Zhang, Xinran Li.

**Supervision:** Tianshu Zhang, Xinran Li.

**Validation:** Tianshu Zhang.

**Visualization:** Tianshu Zhang.

**Writing – original draft:** Zeping Hu.

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
