## [Decision Letter · Decision Letter 0]

18 Nov 2024

Dear Dr. Zhang,

Thank you for submitting your manuscript to PLOS ONE. After careful consideration, we feel that it has merit but does not fully meet PLOS ONE’s publication criteria as it currently stands. Therefore, we invite you to submit a revised version of the manuscript that addresses the points raised during the review process.

We look forward to receiving your revised manuscript.

Kind regards,

Ricardo Jorge Alcobia Granja Rodrigues, Ph.D.

Academic Editor

PLOS ONE

Journal Requirements:

5. Please remove your figures from within your manuscript file, leaving only the individual TIFF/EPS image files, uploaded separately. These will be automatically included in the reviewers’ PDF

Additional Editor Comments (if provided):

Your manuscript has a number of strengths, but before being considered for publication I would recommend that you clearly engage with the suggestions made by reviewer 1.

Reviewers' comments:

Reviewer's Responses to Questions

**Comments to the Author**

1. Is the manuscript technically sound, and do the data support the conclusions?

Reviewer #1: Yes

Reviewer #2: Yes

2. Has the statistical analysis been performed appropriately and rigorously?

Reviewer #1: Yes

Reviewer #2: Yes

3. Have the authors made all data underlying the findings in their manuscript fully available?

Reviewer #1: Yes

Reviewer #2: Yes

4. Is the manuscript presented in an intelligible fashion and written in standard English?

Reviewer #1: Yes

Reviewer #2: Yes

Reviewer #1: This study used CHARLS data to investigate the impacts of long-term care insurance on household expenditure in China. I have the following comments:

Introduction and literature

1. 'As physical functions decline with age, the probability of elderly individuals suffering from chronic diseases rise to 56%'. It is unclear what this sentence means exactly.

2. The concepts of medical care and long-term care should be clearly defined. It is mentioned that 'the aim (of long term care insurance) is to provide comprehensive medical care servies and financial support for the disabled elderly. What are the relationships between long term care and medical care?

3.References should be added to support some of the discussion points. For example, (1) Barthel Scale (2) The description of the healthcare insurance and long-term care insurance in China; (3) previous studies focusing on the impacts of LTCI

Method:

1. Clarify the relationships between health care expenditure, medical expenditure, and non-medical expenditure. Are they mutually exclusive?

2. Table 2: Why the mean value of the LTCI particpation variable for the treated group is 0.5 not 1? And why its standard deviation is also 0.5?

3. Report the level of missingness for the expenditure variables and how missingness has been handled.

4. Equations (2)-(4). The interaction term between Treat and Time is included in the models but the main effects (i.e., Treat and Time separately) are excluded from the models. It is unknown what the interaction term is trying to estimate in this case.

5. In equation (4), Y in time point t+1 is treated as the outcome variable, but Y in time point t is not controlled for. This leads to endogeneity bias.

Results:

1. The discussion relating to LTCI as a health protection tool (above Table 7) should be moved to the discussion section, as these are interpretation of the results.

2. The statistical significance of the mediation effect should be formally tested. Relying on the significance of the two coefficients (i.e., DID on mediator and mediator on final outcome) is not always reliable.

Discussion:

1. 'LTCI reduces the need for preventative savings, which leads to improved wellbeing and quality of life'. These are interpretation of the results and implications of the research findings. Author discussed them in a way as if they are research findings directly supported by data, which can be confusing. These points should be re-written.

2. Authors recommended a care model that combines community and family care and reduce institutional care. Information technology should be integrated. It is difficult to see how these are relevant to the research findings of LTCI in this study.

Reviewer #2: 1. Technical Soundness and Data Support: The study applies a DID approach using longitudinal data from CHARLS, providing empirical support for assessing the impact of LTCI on elderly household expenditures. The data spans three periods (2015-2020), offering sufficient variation for analysis. The conclusions align well with the data findings, which reveal significant effects of LTCI on various types of expenditures for the elderly, supporting the authors' assertions about LTCI's impact on expenditure patterns.

2. Statistical Analysis: Statistical rigor is achieved through a combination of PSM and DID, addressing potential selection biases and establishing robust comparative groups. Additionally, the manuscript conducts robustness checks via individual placebo tests, price index adjustments, and dependent variable substitutions. These elements collectively validate the impact of LTCI, indicating that the statistical analysis was appropriately conducted.

3. Data Availability: The authors have made the CHARLS dataset accessible through a public repository, with details in the manuscript, meeting the criteria for data transparency. This ensures that the findings are reproducible by others with access to the CHARLS data.

4. Presentation and Language: The manuscript is presented in a clear, intelligible manner, with a structured flow from hypothesis development to methodological details, results, and conclusions. Key terms are defined early, and tables summarize statistical results effectively.

**Do you want your identity to be public for this peer review?** For information about this choice, including consent withdrawal, please see our Privacy Policy

Reviewer #1: No

Reviewer #2: No

---

## [Author Response · Author response to Decision Letter 1]

22 Nov 2024

Replies to editor and reviewers

Dear editor and reviewers:

Thank you for offering us an opportunity to improve the quality of our submitted manuscript. We sincerely appreciate your valuable feedback and suggestions. In response to the comments provided by the editor and reviewers, we have thoroughly revised the manuscript and highlighted all the changes in red for clarity.

We hope the revised manuscript has now met the publication standard of your journal. The following is our specific reply to the reviewer's comments.

Response to Reviewer #1:

Introduction and literature

Comment 1: 'As physical functions decline with age, the probability of elderly individuals suffering from chronic diseases rise to 56%'. It is unclear what this sentence means exactly.

Reply 1�Thank you for your valuable feedback regarding the sentence: "As physical functions decline with age, the probability of elderly individuals suffering from chronic diseases rises to 56%."We acknowledge that the original phrasing may have been unclear. Our intention was to highlight the high prevalence of chronic diseases among elderly individuals due to the natural decline in physical functions with age. To address this, we have revised the sentence in lines 35–38 .Thank you once again for your constructive suggestion, which has helped improve the clarity of our manuscript.

Comment 2: The concepts of medical care and long-term care should be clearly defined. It is mentioned that 'the aim (of long term care insurance) is to provide comprehensive medical care servies and financial support for the disabled elderly. What are the relationships between long term care and medical care?

Reply 2�Thank you for your valuable feedback regarding the need to clearly define the concepts of medical care and long-term care, and to elaborate on their relationship. Based on your suggestion, I have revised the manuscript in lines 64–72 to address these concerns.In the revised text, I have explicitly defined medical care and long-term care, highlighting their distinctions. Medical care primarily focuses on addressing acute health needs, while long-term care emphasizes supporting the daily living needs of individuals with chronic illnesses or disabilities. Additionally, I have clarified how long-term care often integrates certain aspects of medical care to ensure comprehensive and continuous support for the elderly, particularly those with disabilities.

Moreover, the revised content elaborates on how LTCI bridges the gap between acute medical services and the long-term care required for chronic conditions, providing both continuity and holistic protection for the elderly population. These changes aim to provide a more precise explanation of the relationship between the two types of care, as well as LTCI’s role in addressing these needs.

I hope this revision meets your expectations, and I appreciate your insightful comments, which have helped enhance the clarity and depth of the manuscript. Thank you again for your constructive input.

Comment 3: References should be added to support some of the discussion points. For example, (1) Barthel Scale (2) The description of the healthcare insurance and long-term care insurance in China; (3) previous studies focusing on the impacts of LTCI

Reply 3�Thank you for your valuable suggestions. Based on your feedback, I have revised the manuscript and added references to support the discussion points in lines 79, 81, 83, 92, 96, 98, and 99. These references are highly relevant and provide robust evidence to substantiate the arguments, ensuring the discussion is well-supported and comprehensive.I sincerely appreciate your insightful comments, which have greatly improved the quality of the manuscript.

Method

Comment 1: Clarify the relationships between health care expenditure, medical expenditure, and non-medical expenditure. Are they mutually exclusive?

Reply 1�Thank you for your insightful comments. We have carefully revised the manuscript to address your concerns. Specifically, in lines 223–239 and 242–243, we clarified the definitions and scope of these expenditure categories, emphasizing their distinctions and mutual exclusivity.

For example, we now explicitly define medical expenditures (HME) as costs directly related to treatments and medical services, while health care expenditures (HHCE) pertain to spending on preventive and wellness-related activities. Non-medical health care expenditures (HNMHCE) include all other household expenditures unrelated to health or medical services. These revisions aim to provide a clearer and more precise framework for understanding the expenditure categories.

Your feedback has been invaluable in improving the clarity and rigor of our study, and we sincerely appreciate your thoughtful suggestions. If there are additional areas where you believe further improvement is needed, we would be grateful for your guidance.

Comment 2: Table 2: Why the mean value of the LTCI particpation variable for the treated group is 0.5 not 1? And why its standard deviation is also 0.5?

Reply 2�Thank you very much for pointing out this issue regarding the mean value and standard deviation of the LTCI participation variable for the treated group in Table 2. Upon reviewing your insightful comment, we realized that this discrepancy was due to an oversight during data entry. We deeply regret this error and have corrected it accordingly. Your attention to detail has been incredibly valuable in helping us identify and address this mistake. Furthermore, we have conducted a thorough review of the entire manuscript to ensure that no other similar oversights exist. We sincerely appreciate your meticulous review and constructive feedback, which have greatly contributed to improving the quality and accuracy of our work. Your expertise and careful examination are truly inspiring, and we are very grateful for the opportunity to benefit from your guidance.

Comment 3: Report the level of missingness for the expenditure variables and how missingness has been handled.

Reply 3�Thank you for your thoughtful comments. To address your concern, we have carefully reviewed the data and added a detailed explanation in the revised manuscript. Specifically, we calculated the percentage of missing values for each expenditure variable and reported these levels in the text. The relevant changes can be found in lines 243–253 of the manuscript. For the handling of missingness, we adopted listwise deletion to ensure the robustness and reliability of the results. This approach was chosen based on its suitability for our data and the potential impact of missingness on the analysis. We deeply appreciate your attention to this critical aspect, as it has helped us enhance the transparency and rigor of our study. If you have further suggestions or believe additional clarifications are needed, we would be grateful for your guidance.

Comment 4: Equations (2)-(4). The interaction term between Treat and Time is included in the models but the main effects (i.e., Treat and Time separately) are excluded from the models. It is unknown what the interaction term is trying to estimate in this case.

Reply 4�Thank you very much for your insightful comment. We deeply appreciate your attention to this critical aspect of the model specification. In response to your suggestion, we have revised the models to explicitly include the main effects of Treat and Time. While we had already accounted for these variables in our regression analyses to ensure the robustness of the results, we inadvertently omitted them in the originally presented equations. By explicitly incorporating these main effects in the revised equations, the interaction term (Treat × Time) now clearly captures the differential treatment effect over time, while the main effects account for baseline differences between groups (Treat) and the general time trend (Time). This adjustment enhances the clarity, interpretability, and alignment of the model with standard econometric practices. The updated equations have been incorporated into the manuscript. We sincerely thank you for your valuable feedback, which has greatly improved the transparency and rigor of our study. Your careful review has been instrumental in helping us refine our work, and we are truly grateful for your guidance.

Comment 5: In equation (4), Y in time point t+1 is treated as the outcome variable, but Y in time point t is not controlled for. This leads to endogeneity bias.

Reply 5�Thank you very much for pointing out the potential endogeneity issue in Equation (4). We deeply appreciate your meticulous review, which has helped us identify this critical concern. In response to your suggestion, we have revised Equation (4) to include (the lagged dependent variable) as a control variable. By accounting for the model now mitigates potential endogeneity bias arising from unobserved factors that may influence . This adjustment ensures the robustness and validity of our results by controlling for the autocorrelation in the dependent variable. The revised equation has been updated in the manuscript, and we have also provided additional explanations in the methodology section to clarify this modification. If you have further suggestions or insights, we would greatly appreciate your guidance to further enhance the rigor of our analysis.

Results

Comment 1: The discussion relating to LTCI as a health protection tool (above Table 7) should be moved to the discussion section, as these are interpretation of the results.

Reply 1: Thank you for your valuable feedback. In response to your suggestion, we have moved the discussion on LTCI as a health protection tool, which was previously placed above Table 7, to the discussion section of the manuscript. This adjustment ensures that the results section remains focused solely on presenting findings, while the interpretation and implications of these results are appropriately addressed in the discussion section. We appreciate your insightful comment, which has helped us improve the structure and clarity of the manuscript.

Comment 2: The statistical significance of the mediation effect should be formally tested. Relying on the significance of the two coefficients (i.e., DID on mediator and mediator on final outcome) is not always reliable.

Reply 2: Thank you for your constructive comment. In this study, we employed the widely used three-step regression approach to conduct the mediation effect analysis. This method involves three key steps:

1.Testing the significance of the independent variable (DID, or X) on the dependent variables (HME, HHCE, HNMHCE, and THE), as shown in the baseline regression.

2.Assessing the significance of the independent variable (DID) on the mediators (Self-assessed health and Intergenerational economic support, or M).

3.Regressing the dependent variables (Y) on both the independent variable (X) and the mediators (M).

This approach has been extensively validated in the literature. For example, Baron, R. M., & Kenny, D. A. (1986). The moderator-mediator variable distinction in social psychological research: Conceptual, strategic, and statistical considerations. Journal of Personality and Social Psychology, 51(6), 1173–1182. https://doi.org/10.1037/0022-3514.51.6.1173�Zhao, X., Lynch Jr, J. G., & Chen, Q. (2010). Reconsidering Baron and Kenny: Myths and truths about mediation analysis. Journal of Consumer Research, 37(2), 197–206. https://doi.org/10.1086/651257�Hayes, A. F., & Rockwood, N. J. (2020). Regression-based statistical mediation and moderation analysis in clinical research: Observations, recommendations, and implementation. Behaviour Research and Therapy, 117, 104636. https://doi.org/10.1016/j.brat.2019.104636.

To ensure clarity and focus in the manuscript, we included the results for the latter two steps in Tables 9 and 10. However, it is important to note that we also applied the Bootstrap method to formally assess the statistical significance of the mediation effects. The Bootstrap results were consistent with our three-step regression findings, further confirming the robustness of the mediation effects. Considering the space limitations of the manuscript, we did not include the detailed Bootstrap results in the current version. However, if you believe that presenting these results would enhance the transparency and completeness of the analysis, we would be happy to include them in the revised manuscript.

Your insightful comment has encouraged us to carefully reflect on the methodological rigor of our analysis, and we sincerely appreciate the opportunity to further refine our work based on your guidance.

Discussion

Comment 1: 'LTCI reduces the need for preventative savings, which leads to improved wellbeing and quality of life'. These are interpretation of the results and implications of the research findings. Author discussed them in a way as if they are research findings directly supported by data, which can be confusing. These points should be re-written.

Reply 1: Thank you very much for your insightful feedback. We deeply appreciate your careful review and valuable suggestions, which have helped us refine our manuscript. In response to your comment, we have revised the discussion in lines 487–492 to ensure clarity and precision. Specifically, we have rephrased the statements related to LTCI's impact on preventative savings, well-being, and quality of life. The revised text no longer presents these points as direct findings supported by the data but instead discusses them as potential implications derived from the observed effects of LTCI on household expenditures. This change emphasizes the exploratory nature of these implications and avoids any overstatement of the results. Thank you again for highlighting this important aspect, which has significantly improved the rigor and clarity of our discussion. If you have further suggestions or feedback, we would be delighted to address them.

Comment 2: Authors recommended a care model that combines community and family care and reduce institutional care. Information technology should be integrated. It is difficult to see how these are relevant to the research findings of LTCI in this study.

Reply 2: Thank you for your insightful feedback regarding the care model recommendations in the manuscript. To address your comments, the content in lines 541–552 has been revised to better align the proposed recommendations with the research findings on LTCI. The updated discussion highlights that a care model combining community and family care can enhance the effectiveness of LTCI, particularly in rural areas with limited healthcare resources. This approach reduces reliance on traditional institutional care and addresses the specific challenges identified in the study. Additionally, the integration of information technology has been emphasized as a key strategy to improve service efficiency and personalization. We sincerely appreciate your valuable comments, which have significantly enhanced the focus and clarity of the manuscript.

Response to Reviewer #2:

Comment 1: Technical Soundness and Data Support: The study applies a DID approach using longitudinal data from CHARLS, providing empirical support for assessing the impact of LTCI on elderly household expenditures. The data spans three periods (2015-2020), offering sufficient variation for analysis. The conclusions align well with the data findings, which reveal significant effects of LTCI on various types of expenditures for the elderly, supporting the authors' assertions about LTCI's impact on expenditure patterns.

Reply 1�We sincerely thank the reviewer for the positive feedback on the technical soundness and data support of our study, as well as for your valuable comments. We are delighted that you found our application of the DID approach and the use of longitudinal data from CHARLS to be effective in analyzing the impact of LTCI on elderly household expenditures. Your recognition of how our conclusions align with the data findings is especially encouraging. We will carefully consider your comments and continue to refine our work to ensure the study is as rigorous and valuable as possible.

Comment 2: Statistical Analysis: Statistical rigor is achieved through a combi

---

## [Decision Letter · Decision Letter 1]

16 Dec 2024

The Impact of Long-Term Care Insurance on Household Expenditures of the Elderly: Evidence from China

PONE-D-24-29983R1

Dear Dr. Zhang,

We’re pleased to inform you that your manuscript has been judged scientifically suitable for publication and will be formally accepted for publication once it meets all outstanding technical requirements.

Kind regards,

Ricardo Jorge Alcobia Granja Rodrigues, Ph.D.

Academic Editor

PLOS ONE

Additional Editor Comments (optional):

Reviewers' comments:

Reviewer's Responses to Questions

**Comments to the Author**

Reviewer #1: All comments have been addressed

2. Is the manuscript technically sound, and do the data support the conclusions?

Reviewer #1: Yes

3. Has the statistical analysis been performed appropriately and rigorously?

Reviewer #1: Yes

4. Have the authors made all data underlying the findings in their manuscript fully available?

Reviewer #1: Yes

5. Is the manuscript presented in an intelligible fashion and written in standard English?

Reviewer #1: Yes

Reviewer #1: Authors have addressed all of my comments well. In particular, I am glad that authors have used the bootstrap method to formally assess the statistical significance of the mediation effects. I am aware that the Baron-Kenny's three-step approach has a long history and has been widely used in previous studies. But there has also been an increasing number of studies pointing out its flaws in reasoning: having two statistically significant coefficients does not guarantee that the product of the two coefficients is statistically significant as well (sometimes it does while other times it does not). It has always been my view that formally testing the mediation effects will definitely help to further enhance the rigour of a study.

**Do you want your identity to be public for this peer review?** For information about this choice, including consent withdrawal, please see our Privacy Policy

Reviewer #1: No

---

## [Editor Report · Acceptance letter]

PONE-D-24-29983R1

PLOS ONE

Dear Dr. Zhang,

I'm pleased to inform you that your manuscript has been deemed suitable for publication in PLOS ONE. Congratulations! Your manuscript is now being handed over to our production team.

Kind regards,

on behalf of

Dr. Ricardo Jorge Alcobia Granja Rodrigues

Academic Editor

PLOS ONE